# Exploration of the impact of the COVID-19 pandemic on people with dementia and carers from black and minority ethnic groups

Emily West ,[1] Pushpa Nair,[2] Yolanda Barrado-Martin,[2] Kate R Walters,[2] Nuriye Kupeli,[1] Elizabeth L Sampson ,[1,3] Nathan Davies [2]

► Prepublication history and supplemental material for this paper is available online. To view these files, please visit the journal online (http://dx.doi.org/10.1136/bmjopen-2021-050066).

EW and PN contributed equally.

EW and PN are joint first authors.

[1]Marie Curie Palliative Care Research Department, Division of Psychiatry, University College London, London, UK
[2]Department of Primary Care and Population Health, University College London, London, UK
[3]Division of Psychiatry, University College London, London, UK

**Correspondence to**
Dr Emily West;
emily.west@ucl.ac.uk

## ABSTRACT

**Introduction** Despite community efforts to support and enable older and vulnerable people during the COVID-19 pandemic, many people with dementia and their family carers are still finding it difficult to adjust their daily living in light of the disruption that the pandemic has caused. There may be needs specific to black, Asian and minority ethnic (BAME) populations in these circumstances that remain thus far unexplored.

**Objective** The aim of the study was to explore the effects of the COVID-19 pandemic on people living with dementia and their family carers of BAME backgrounds, in relation to their experiences of community dementia care and the impact on their daily lives.

**Design** 15 participants (persons with dementia and carers) were recruited for semistructured qualitative interviews. Respondents were of South Asian and Afro-Caribbean backgrounds. We used thematic analysis to analyse our data from a constructivist perspective, which emphasises the importance of multiple perspectives, contexts and values.

**Results** There were a number of ways that the COVID-19 pandemic has impacted BAME persons with dementia and carers with regard to their experiences of dementia community care and the impact on their everyday lives. In particular we identified eight key themes, with subthemes: fear and anxiety, food and eating (encompassing food shopping and eating patterns), isolation and identity, community and social relationships, adapting to COVID-19, social isolation and support structures, and medical interactions. Fear and anxiety formed an overarching theme that encompassed all others.

**Discussion** This paper covers unique and underexplored topics in a COVID-19-vulnerable group. There is limited work with these groups in the UK and this is especially true in COVID-19. The results showed that such impacts were far-reaching and affected not only day-to-day concerns, but also care decisions with long-ranging consequences, and existential interests around fear, faith, death and identity.

## BACKGROUND

Over 850000 people in the UK have dementia and an estimated 25000 of these come from black, Asian and minority ethnic (BAME) groups, defined in the UK as non-white British.[1]

### Strengths and limitations of this study

► This paper covers underexplored topics among at-risk groups during COVID-19: black, Asian and minority ethnic groups are both particularly vulnerable and understudied in this context.
► Interviews were undertaken using Microsoft Teams and online videoconferencing platform.
► This study focused on South Asian and Afro-Caribbean groups and views may not be generalisable to other minority groups.
► Data were analysed using an iterative constructivist approach and were coded thematically.
► Analysis was performed by a diverse team.

The National Audit Office[2] has proposed that special dementia services may be needed to support these groups. There is evidence that people from ethnic minority backgrounds seek less support from services,[3] possibly as a result of increased stigma, alternative health beliefs, reduced knowledge about dementia and perceived lack of culturally tailored care.[3–6] There is also evidence that certain minority backgrounds may be less likely to receive a diagnosis of dementia despite increased risk.[7 8] There is often reluctance among BAME communities to enter residential or nursing care, with most care being undertaken by family members in the community.[9] This may be due to cultural norms and beliefs about caring for older family members and multigenerational living already being more common. Fears about culturally sensitive or appropriate care may also come into play. Indeed, the majority of people living with dementia live in the community in their own homes or with their family.[10] In the UK, there are currently around 700000 family members and friends acting as primary carers (ie, providing the majority of care).[11]

Despite community efforts to support and enable older and vulnerable people during

the COVID-19 pandemic, many people with dementia and their family carers are still finding it difficult to adjust their daily living in light of the disruption that the pandemic has caused.[12] There may be needs specific to BAME populations in these circumstances that remain thus far unexplored.

Aspects of dementia care such as caring arrangements and access to health and social care have been widely affected. We wanted to explore the effects of the pandemic on general well-being of both people with dementia and carers from BAME communities, particularly in light of raised concerns over the higher incidence of COVID-19 in BAME groups and higher risk of complications.[13] BAME communities have also been found to be particularly COVID-19-vulnerable, with people of a Bangladeshi background being twice as likely to die from COVID-19 as those of a white British background. Similarly, people of a black Caribbean or other black backgrounds are at 10%–50% increased risk of death compared with white British.[14] Older adults of BAME background have also expressed that they are more likely to decline COVID-19 vaccine than white Britons,[15 16] which suggests that this gap may yet widen further. In light of these facts, this study concentrates on BAME community members of South Asian and Afro-Caribbean backgrounds to help us understand what the real challenges are in order to develop ways to promote tailored support throughout the pandemic. This study focuses on community care, as this model of help-seeking was heavily promoted to caregivers as pressure on acute services increased.

The aim of the study was to explore the effects of the COVID-19 pandemic on people living with dementia and their family carers of BAME backgrounds, in relation to their experiences of community dementia care and the impact on their daily lives.

## RESEARCH DESIGN AND METHODS
### Participants
This study was part of the ENDEMIC: dEmeNtia and DEcision MakIng during COVID-19 programme, funded by the Economic and Social Research Council, aimed at developing an evidence-based decision tool to support family carers and people with dementia to make these difficult decisions during COVID-19. We recruited 15 participants in total, 11 family carers and 4 persons living with dementia. Participants were recruited from a variety of sources, including previous studies, online dementia research websites (such as Join Dementia Research), social media and interested relevant local/national carer organisations. Participants were recruited from across the UK, although primarily from in and around Greater London due to the high proportion of BAME communities in this area.

### Inclusion criteria
Inclusion criteria were current carers of South Asian and Afro-Caribbean backgrounds providing unpaid care to someone living with dementia, or people over the age of 65 with dementia of South Asian or Afro-Caribbean background who have dementia but with the capacity to consent.

### Exclusion criteria
Exclusion criteria were carers with cognitive impairment or bereaved carers; people with dementia diagnosed under 6 months from interview date; participants who do not have mental capacity to provide informed consent; and participants who were not of South Asian or Afro-Caribbean backgrounds.

### Recruitment
Participants recruited by local/national dementia/carer networks were initially approached by the organisation; this was done in line with their local organisation policies and guidelines and included email, newsletters and direct contact. Participants were also approached directly by the research team. These participants were identified through past involvement in studies and ongoing permission to contact, and contact was initially through email or telephone. All interested participants were asked to contact the study team directly, or gave permission to be sent a copy of the study advert, study information leaflet and consent form (according to the participant's preferred method of communication—via email or post). Interested participants were encouraged to contact the research team via email or telephone and were then all telephone-screened against the study criteria and given an opportunity to ask questions.

Consent forms were completed electronically and sent back to the research team via email, where possible. They were then countersigned by the research team and returned to the participant via email, to keep for their records. Special accessible consent forms were made available for people with dementia. If participants were unable to complete the consent form electronically, consent was taken verbally and audio-recorded prior to the start of the interview. A hard copy of the consent form was then filled in, signed by the researcher and emailed or posted back to the participant, to keep for their records.

### Patient and public involvement
A patient and public involvement (PPI) panel was consulted throughout the research process. The panel provided input on devising the research question, leading us to focus on community care and experience. The panel also gave feedback on the design of the study—namely managing the burden of interviews to persons with dementia and carers—formulation of the topic guide and interpretation of findings.

### Data collection
We collected data through individual semistructured interviews, conducted remotely over telephone (n=13) or via secure video technologies (Microsoft Teams, n=2). Interviews were conducted by PN (female) (n=14; an academic general practitioner (GP)) and EW (female)

(n=1; a researcher). Interviews lasted approximately 1 hour (range: 35–75 min) and were audio-recorded using a Dictaphone. The interview topic guide was developed with the input of the research team, including our PPI representatives, and was modified as interviews progressed. The topic guide (online supplemental appendix 1) explored the impact of the COVID-19 pandemic on BAME carers and people with dementia, considering daily routines, eating and drinking, well-being, access to dementia services, healthcare and social care, and exploring what extra support they would have found useful.

A distress protocol was devised in case participants became distressed during interviews, but we had no recourse to use this. All participants were able to converse in English, although proficiency levels varied. Both interviewers have experience in working internationally and with persons with communication difficulties. No extra formal interventions were required to facilitate interviews.

At the end of each interview, we collected demographic data on age, gender, marital status, ethnic group, country of birth, first language, education level and work. Audio-recorded interview data were transcribed verbatim, anonymised and verified for accuracy by the interviewer.

### Data analysis

We used inductive thematic analysis to analyse our data from a constructivist perspective, which emphasises the importance of multiple perspectives, contexts and values.[17] Our team included two academic GPs (PN, KRW), a senior qualitative researcher (ND) and two early career researchers (EW, YB-M). PN developed a thematic framework, which was further refined with input from YB-M, EW and ND. This framework was further refined by EW. EW then iteratively coded the transcripts using NVivo V.11 according to the thematic framework and in discussion with the wider research team.

### RESULTS

The demographic characteristics of the participants are detailed in table 1. In interviews, eight key themes, with subthemes, were identified: fear and anxiety, food and eating (encompassing food shopping and eating patterns), isolation and identity, community and social relationships, adapting to COVID-19, social isolation and support structures, and medical interactions.

### Fear and anxiety

It makes me worried, but at the same time, I accept that it is all over the world. Nobody will escape this corona. So, I accept it as God's will. I don't know what it is. —PwD 01 (Male, 88)

Fear and anxiety as a theme was discussed both explicitly and was reflected implicitly throughout other themes. This topic proved to be an overarching theme that was a foundation for other subjects discussed. Fear took many forms, from overarching existential fear surrounding the

| Table 1 | Demographic characteristics of participants | |
|---|---|---|
| **Demographics** | **PwD, n=4 (%)** | **Carers, n=11 (%)** |
| **Age** | | |
| Under 50 | 0 | 9 |
| 50–59 | 0 | 36 |
| 60–69 | 25 | 9 |
| 70–79 | 50 | 27 |
| 80+ | 25 | 18 |
| **Gender** | | |
| Male | 50 | 9 |
| Female | 50 | 91 |
| **Marital status** | | |
| Single | 25 | 27 |
| Married | 25 | 64 |
| Divorced | 25 | 0 |
| Widowed | 25 | 9 |
| **Ethnic group** | | |
| Caribbean | 50 | 45 |
| Indian | 50 | 45 |
| Pakistani | 0 | 9 |
| **Country of birth** | | |
| India | 25 | 27 |
| Jamaica | 50 | 0 |
| Barbados | 0 | 9 |
| Kenya | 0 | 9 |
| Trinidad and Tobago | 0 | 9 |
| Uganda | 25 | 9 |
| UK | | 36 |
| **First language** | | |
| English | 50 | 45 |
| Indian languages (Bengali, Gujarati, Punjabi, Urdu) | 50 | 54 |
| **Age left education** | | |
| Age 17–20 | 0 | 36 |
| Over 20 | 100 | 64 |

PwD, persons with dementia.

concept of risk or serious illness or death, to anxieties about specific situations or susceptibilities, such as risks inherent in using public transport, going to work or seeing family. Many fears were expressed in conversations about comorbidities—factors that made the carer or person with dementia more susceptible to contracting COVID-19 or less likely to respond well to treatment. There was some awareness in the group of respondents that BAME individuals were at generally higher risk due to being over-represented in front-line jobs. This was sometimes attributed to having more limited work options available. This was an additional form of anxiety for many, as was the worry that they could bring the virus into contact with those with dementia by virtue of these roles.

Yes, I do, yes, because I think our people get the worst of all worlds. And you know, we may say, well, if you don't like the job get yourself another job. But where are you going to get it? And if you can get it, who is going to give it to you? This is it. —PwD 02 (Female, 75)

## Food and eating

One of the most discussed subjects among carers was food and eating, which was specifically explored in interviews. Many issues around food were discussed and were split into two main themes: shopping for food and eating food.

## Food shopping

Food shopping was a cause of some stress for many carers, specifically in the early stages of the first lockdown where rules around priority access were not fully in place. There was also considerable anxiety around the safety of shopping for food, with fears around exposing the person with dementia to COVID-19 through lack of social distancing in supermarkets or bringing the virus into the home on food. Many carers reported that they sanitised food bought in supermarkets to mitigate this risk. These anxieties made food shopping a time-consuming and fraught process for carers. Similarly, a number of persons with dementia expressed frustration that they no longer had the independence to go to supermarkets and pick out what they wanted due to shielding.

But the queues, I had to go to [supermarket] … You don't get it so much now, but the queues were a mile long. You didn't know when you went shopping how long the queue was to get in. All of that. —Carer 10 (Female, 57)

## Eating patterns

Changes to food behaviours—defined widely as behaviours that encompass food shopping, preparation and mealtimes—brought on by the pandemic were also reflected in eating patterns. It was widely reported both by our respondents and in the media that the food parcels given out by councils to vulnerable members of the community (which included many of our sample) were not ethnically targeted or sometimes even appropriate. For example, respondents following specific diets—vegetarian and halal—received many items that they could not eat. A number of Caribbean respondents reported that culturally common foods such as plantain and yams were not available to them, especially early in the pandemic.

Yes, I would have thought so. Basically, people of their own culture, like Indians, we would prefer to have obviously Indian food. That doesn't mean that we don't like Western food or Chinese food, for that matter. Of course. That is there. I wouldn't refuse chole puri [a chickpea dish] [any day]. —Carer 09 (Female, 72)

Some respondents reported that they adapted their diet based on foods available or given to them and ate foods that they had not before. A number of persons with dementia also reported temporarily cutting unavailable foods out of their diets, rather than going to greater lengths to secure access to them.

Yes but also the tinned food. I never used to eat things like that and I used to eat the right food, fresh fruit and fresh veg and I love fish. Fish and stuff. And it's just like tins of tuna. Since the lockdown I've had two corned beef and it's not that bad when I eat it, but for my age, so I've got to watch what I'm doing but I had no choice. —PwD 04 (Female, 66)

Some eating-focused adaptations to the pandemic were positive, with carers and person with dementia reporting that diets had improved or become healthier in this time due to having more time to make fresh meals. Carers often felt reassured that they could better monitor the person with dementia's food intake and dietary habits. Being able to express care through cooking, sharing food and ensuring nutritious food that was likely to be appealing to the person with dementia was seen as positive. However, the extra work and worry associated with being responsible for the person with dementia's food intake was often seen as a negative.

So it was a heavy impact on me and I think again it's that being pulled in two ways. Being worried about myself as an individual, but actually this is my job and I've got to do it before anybody says anything. —Carer 07 (Female, 57)

## Isolation and identity

All respondents interviewed talked about their own wellbeing and how this has changed over the course of the pandemic. Many carers reported feeling strained, without their usual outlets and opportunities to have an identity outside of being a carer. A loss of freedom was particularly reported among older spousal carers, as they also often had comorbidities that put them at risk. This made some couples heavily dependent on outside or familial help for tasks that they had previously been able to manage independently. Reduced social interaction was widely reported by both groups of respondents, and often behind this was a fear of contracting COVID-19 and transmitting this to loved ones. This often led carers to report that they adhered to self-imposed lockdowns more strict than those imposed by the government.

Just that carers feel isolated, even before the lockdown. So, in the lockdown there's a lot of pressure to follow the guidelines, to do everything with the limitations. Things like going to the grocery [store], to buy groceries is extremely stressful because having to get everything so you don't have to keep going out all the time. —Carer 04 (Female, 29)

As with all themes, experiences of isolation and redefining identities were very individualistic, so the downsides of a situation for some proved to be upsides for others. For some this was related to financial savings, or having

the ability and support to access online services and activities. For example, a number of creative solutions to cancelled activities and restrictions were reported, from going for an evening drive to attending Zoom worship services or online exercise classes. However, for some this took the form of a relaxing of social pressures. One spousal carer reported that lockdown actually enabled her to rest as she could no longer go out and run errands. In many cases there was also more family help available and greater financial support able to be accessed. This was largely related to individual or family circumstances than culture per se.

> I've got into gardening even more. I did a course in horticulture last year, and that has helped. So, I've just gone out and bought loads of plants. It's amazing actually. A lot of people have said this, but just buying a whole lot of plants and actually learning how to care for these plants. —Carer 10 (F, 57)

### Community and social relationships

Religion and cultural celebrations were important for many respondents, and being unable to go to church, temple or similar was very difficult for some. Many holidays, observances and celebrations were missed, and these were usually considered very communal experiences. In their place, personal prayer gained heightened importance. Rites of passage were also affected by COVID-19, with weddings and baptisms being postponed and cancelled. Funerals were also widely affected, and this had particular significance in cultures where seeing the body or burial within a certain timeframe is important. Similarly, having a large well-attended funeral is seen as a mark of respect in many communities, and it was upsetting to the respondents that this was not happening during the pandemic when community elders died.

> That's quite hard. In terms of traditionally, that's what we do. You have to say your goodbyes. And if you can't, that's the worst possible thing, really, if you can't go to somebody's funeral. —Carer 01 (Female, 62)

Social dynamics were discussed by many respondents. Many carers reported that being distanced from their children and extended family—particularly early in the pandemic where tight restrictions on interactions had been applied during the first lockdown—contributed to their feelings of isolation and increased workload. Suddenly jobs which had previously been shared across a whole (extended) family now fell to one or two members, especially if some family members had high-risk jobs that put them in contact with many members of the public. Many older carers also reported that their children felt stressed about the risk of transmitting COVID-19 to older parents and so reduced contact. This led to deeper feelings of social isolation.

> Do you know what? We just prayed. Because we were being so safe, wearing gloves ourselves and putting slippers on. And me and my sister said to each other, if any of us became ill, we wouldn't go. The other one would carry the burden. So, we just prayed. We didn't think about it, actually. We just hoped it wouldn't happen. —Carer 06 (Female, 57)

There were some positive adaptations to these issues, with people reporting that they had learnt to use new devices or forms of technology such as Zoom, FaceTime and monitoring technologies for the first time. Some carers reported that they had encouraged or helped older relatives to use technology during the lockdown despite being unable to before, and that this had had a positive effect on their family and social dynamics.

### Adapting to COVID-19

Both persons with dementia and carers were asked about their awareness of, and attitudes towards, COVID-19 during interviews. Most persons with dementia were reported by carers to have an awareness of their own higher risk due to COVID-19 susceptibility. Many also self-reported that they generally accepted the situation with a sense of things being outside of their control, 'what will be will be':

> I don't care about coronavirus because a lot depends on God. —PwD 03 (Male, 76)

When persons with dementia reported their own awareness of COVID-19, they either understood the situation well or expressed an understanding of danger without the specificities of understanding the risk profile of an airborne pandemic virus. For example, there was generally a good understanding of the fact that usual life had shut down, shops were closed, etc, but this was contrasted with continuing frustration over not being able to go out, see friends and partake in normal activities and routines.

> Because I've accepted it is something we cannot do anything about, and it's just not me. It's everybody now, everybody. That's the first time, things like that, where we are… The equality is really meant what it really says. We're all equal now. So, I don't worry too much because it's not me alone, it's everybody. —PwD 02 (Female, 75)

Behavioural changes in persons with dementia, such as increasing depression, anxiety, aggression and sleeplessness, were reported by carers (who tended to be caring for people with more advanced dementia than the persons with dementia that we interviewed directly), and this was more often reported in persons with dementia who did not have good awareness of the reality of COVID-19. For these individuals, grasping aspects of changed rules and environments, such as social distancing, was difficult or impossible, and this in turn was a source of stress for carers. Heightened anxiety was reported by carers in persons with dementia who both understood and did not

understand the full concept of the COVID-19 pandemic. Carers of persons with dementia who had the capacity to modify their own behaviour in relation to COVID-19 risk reported less of this type of anxiety. Carers of persons with dementia who did not practise such behaviour modification ascribed behavioural changes to lack of routine and social interaction, and this was also associated with greater incidence of behaviours-that-challenge such as sleep/wake disturbance, aggression and depression.

> Because often they will blame you for everything or be very aggressive. Especially when tensions are high and you're doing everything but still getting blamed and aggression directed towards you. And I think it's very easy for carers' mental health to decline very quickly in lockdown. —Carer 04 (Female, 29)

### Social isolation and support structures

Different behaviours in accessing support structures were often related to burden. Many caregivers reported that they cancelled the services of paid carers or ancillary staff such as cleaners during the pandemic due to fear of exposing the person with dementia, which increased the sense of burden on partners and family members.

> Coronavirus didn't bother me. Only for him. I stopped the carers for him. I stopped going for walks because of him, because he's at very high risk even now. But it didn't bother me. I said, if it has to come, it'll come. And if I have to die, I have to die. It didn't bother me at all. But its impact on life was changed, that we couldn't do things that we were doing. —Carer 02 (Female, 71)

In addition to this, almost all respondents reported that community dementia care services were largely absent during lockdown. This was acutely felt by both persons with dementia and carers. Before the pandemic, services like dementia hubs or day centres would provide structure to the week and allow respite time for carers. The absence of these services meant that carers struggled to find time for self-care and other activities outside of the person with dementia's capacity, such as supermarket shopping. Persons with dementia also felt this loss, as their social interactions and activities were significantly lessened without them. Many of these services were reported to have started up again virtually as lockdown progressed, but many carers and persons with dementia reported finding it difficult to engage by this medium and experienced a steep learning curve in setting it up. Those who did not have strong links with these types of services before the pandemic reported that they could not access them as a new service user during lockdown. A strong existing bond was seen to be key to accessing this particular form of help.

> It's been terrible, because normally my mum would go on Dial-a-ride to the gurdwara, once a week. She'd go every Wednesday. And, obviously, the gurdwaras are closed. Because she used to go once a week and see all her friends, and that's been a big impact. And also, the day centres are all closed, and that has had an impact I think on her, because she's been quite isolated. I think she's been a bit lonely. —Carer 06 (Female, 57)

Access to support services in general was location-dependent. Some carers and persons with dementia reported having good access to supportive services such as carers, support lines and community support, whereas others reported having none. Although many appreciated the community spirit that precipitated neighbours and local groups offering help, some were wary of the lack of regulation and safeguarding surrounding such services delivered to vulnerable people. Discussion of differences in service provision often precipitated discussion of government handling of COVID-19. Although many were happy with communication and rules—and often phrased this in terms of appreciating that it was a difficult job in an uncertain situation—some were critical of the lack of clarity and provision provided by both central and local governments.

> But the government advice has been so ambiguous and changing. I feel like the lockdown didn't happen soon enough and I feel like the restrictions are being eased too quickly. —Carer 04 (Female, 29)

### Medical interactions

Many respondents had had interactions with medical services during the pandemic, whether as an inpatient or an outpatient. The majority of respondents had only outpatient medical interactions in this time. Most reported having good access to GP surgeries, although they did report missing face-to-face contact with their provider. In some cases, especially later in the pandemic, GPs were more flexible with inperson appointments and this was widely appreciated by the respondents. Services allied to medicine—for example, memory services, podiatry, hearing aid services and eye tests—were widely unavailable during the pandemic, and respondents reported falling behind on regularly scheduled care and being anxious about this. This was also frequently reported regarding hospital appointments.

> You can't just pick up and go to the GP and say, this is a problem or that. So, it felt like a bit of a fight, those everyday things. It just added to it. —Carer 10 (Female, 57)

There were a wide range of experiences related to hospital care. In the UK, hospital visiting has not been allowed during COVID-19. All respondents who had experienced the person with dementia requiring inpatient care reported that this was particularly disorienting for the person with dementia due to not being permitted to have family or other support present during admission. For carers, this was especially stressful when they felt that

communication with the medical team was impaired. Some respondents felt 'left-out' or unable to access information about the care or prognosis of the person with dementia. However, some reported exceptional engagement with hospital staff and appreciative of the way that they facilitated the person with dementia to access communications technology during their admission.

> It was quite traumatising for me as well, because I wasn't there next to her, and I just didn't know what the hell was going on. Waiting for the doctor to give me a call after they had visits and so on. It wasn't pleasant at all. —Carer 09 (Male, 72)

### Planning and decision-making

Most carers reported that they had not had care-planning conversations with the person with dementia at the beginning of the pandemic and that decisions (when required) were made in collaboration with the medical team and extended family. Being confronted with the need to make these decisions in a time of heightened risk and lessened service availability was a key source of stress for many. Persons with dementia who were interviewed directly tended to express a similar 'decisions will be made when they need to be' outlook.

> My only concern was that if anything happened to him, I didn't want him to go to the hospital. I was scared of this Corona. —Carer 08 (Female, 85)

Decisions were typically made with the significance of new regulations—for example, on visiting, contact and triage—at the forefront of the decision-making process. This meant choosing to forgo certain regular treatments even when services opened back up, as well as choosing home care over hospital or care home admittance, or limiting the involvement of outside carers.

### DISCUSSION

This paper covers unique and underexplored topics in a COVID-19-vulnerable group. Focused research on the needs of persons or groups who are often disadvantaged in an emergency health situation is very much needed. There is limited work with these groups in the UK and this is especially true in COVID-19. Themes were wide-ranging and identified a number of ways that the COVID-19 pandemic has impacted BAME persons with dementia and carers with regard to their experiences of dementia community care and the impact on their everyday lives. The results showed that such impacts were far-reaching and affected not only day-to-day concerns but also care decisions with long-ranging consequences, and existential interests around fear, faith, death and identity.

### Support or a lack thereof

BAME groups have been disproportionately affected by the COVID-19 pandemic in the UK.[18] In conjunction with this, a previous meta-analysis has identified significant barriers faced by members of BAME groups in accessing dementia care[5] both at the community and at the service level. Many of our respondents, both persons with dementia and carers, spoke about their own experiences of help-seeking—whether in light of a lack of available services or difficulty in the process of seeking them. Uncertainty around service provision, lack of access without prepandemic relationships with services and practical difficulties in accessing support digitally were all sources of anxiety. Some patients and caregivers will not have these relationships as the pandemic has changed their needs and thus required support levels. Independently of COVID-19, some people's needs will also naturally have changed due to decline over time.

Both persons with dementia and carers remarked on feeling great loss at a lack of routine and identity outside of the caring relationship. Previous research[19] has shown strong correlation between carer burden and subsequent depression and well-being. This has been shown to be related to use of outside sources of support. In a circumstance where outside support has been found to be difficult to access across the board, particular attention should be paid to the burden of burnout and depression in carers moving forward. Similarly, in persons with dementia independence and autonomy have been posited as key to maintaining positive identities in bound circumstances, namely care homes.[20] These will have been affected by COVID-19 at both the micro and macro levels, from government regulations to changes to routines made at the family level in response to fear and anxiety.

A key form of support that was specifically probed in interviews was behaviour around food and eating. Food insecurity throughout the pandemic has been widespread and related to external factors such as employment security, stability of housing and isolation. Barker and Russell[21] identify a number of groups who were particularly at risk of food insecurity under the first COVID-19 lockdown, of which our respondents typically occupy at least two, namely being members of minority populations and having or being a carer for somebody with a chronic illness. Thus, many respondents had been recipients of government-supplied food parcels, which were standardised and tailored towards a typically white British diet. Food and eating behaviours are inextricably tied up with ethnic and cultural identity, and the lack of availability of culturally appropriate food and ingredients during the first COVID-19 lockdown was the source of some distress for a number of interviewees.

This concern was also echoed in reference to community support and social interaction. Many reported a lack of being able to engage with usual forms of rites of passage, cultural milestones or celebrations with their community, and this in turn added to feelings of isolation and lack of identity. Often, attendance at rites of passage, particularly funerals, signifies a person's social standing within the community. Thus the impact of having a sparsely attended funeral is upsetting for those left behind and may contribute to compounded and complex grief.[22]

These factors, in conjunction, constitute a form of double disadvantaging whereby members of BAME communities living with or caring for those with dementia are not only disproportionately affected by the COVID-19 pandemic through systemic deprivation, but also by known barriers to service access, support or disenfranchisement through lack of access to culturally and ethnically appropriate provision, genetic factors and over-representation in front-line occupations.

### Making plans and decisions

The majority of respondents, both persons with dementia and carers, had not made key medical or care decisions in advance of the COVID-19 pandemic. Previous research suggests that there are several reasons for such conversations not taking place in advance, from an expectation that family members will decide what is best[23] to a lack of tailored interventions that positively promote advance decision-making in minority groups.[24] Many carers reported feeling overwhelmed by information in the media about the level of risk and potential lack of provision for older people regarding medical care. A continued cause for concern was the restriction of visitors in acute medical and tertiary care settings. Several interviewees, both persons with dementia and carers, remarked that medical interactions had been especially stressful in light of a person with diminished capacity not having a family member to help them communicate and advocate in the medical setting. A COVID-19-prompted rapid review of evidence around decision-making in older adults[25] identified the need for interventions to be culturally tailored and that several studies highlighted a lack of positive response among BAME communities to generic planning interventions. Data from the interviews in this paper support these findings, from differing familial roles in decision-making to the need to recognise culturally specific ideas around autonomy and 'what is proper'.

### Adapting to COVID-19

Adapting to new restrictions and ways of living under the first COVID-19 lockdown in the UK proved to be challenging for many, but those living with dementia and those involved in caring for persons with dementia faced extra challenges around care provision, help-seeking and lack of service availability. This in some cases is also compounded by difficulties in learning the skills required to access services remotely. Giebel *et al*[26] identified a significant reduction in support service usage among a population of persons with dementia and their carers in the general population. This has potential significant future effects on distress and mental well-being levels in both persons with dementia and carers. Burden of behavioural and psychological symptoms of dementia (BPSD) has been consistently found to be associated with an increased risk of nursing home admission.[27] Thus, the increase in those dealing with BPSD in persons with dementia under lockdown may prove to have significant repercussions on

public health through potential increase in future care home admissions.

Technology was used to access a huge range of services and interactions within our respondent group, from healthcare to social and religious ceremonies. As our respondents in a number of cases were recruited from an online dementia research registry, our sample may skew more tech-savvy than most. Many outreach groups strongly promoted digital literacy training during the pandemic, which may also have had a positive effect. However, those without the capacity to access digital services may end up further disadvantaged. Previous literature[28 29] has highlighted the importance of enabling older adults with dementia and carers to use technology to promote social interaction, and the unique circumstances of the COVID-19 pandemic may have hastened the need for widespread digital inclusion—both for social needs and wider functions, such as maintaining contact with those in hospitals and care homes, attending telehealth appointments and ordering groceries.

### COVID-19 and dementia

Emerging evidence suggests that the impact of COVID-19 on persons living with dementia and their families has highlighted specific challenges and support needs. Many are concerned that the effects of lockdown—lack of social contact, decreased cognitive stimulation and lack of outside purpose—may have a lasting and deleterious effect on those with dementia and other cognitive impairments.[30] These findings were also echoed in the experiences of our respondents. Studies that have measured the psychological impact of pandemic in this population have found increased levels of depression, anxiety and loneliness,[31] which were particularly related to a lack of understanding and knowledge about the pandemic and needs for lockdown. A national longitudinal survey of support service usage and mental well-being in persons living with dementia and their families[32] found a significant drop in usage shortly after lockdown was enacted. This is an experience shared by the participants in this study, who expressed many difficulties, fears and anxieties in accessing services during this time. This study adds a valuable BAME-specific point of view to current literature.

### Strengths and limitations

This paper covers underexplored topics among at-risk groups during COVID-19. BAME groups are both particularly vulnerable and understudied in this context. However, there are certain limitations to our approach. This study focused on South Asian and Afro-Caribbean groups and views may not be generalisable to other minority groups. However, the spirit of the findings— that is, ensuring cultural appropriateness in services and interventions—is widely applicable. We also had a likely self-selecting tech-savvy study population. Due to some aspects of recruitment (via online registries) and the mode of study (interview via Zoom or other video call), we reached people who already had a decent level of

access to technology, and/or could be facilitated in this by an informal carer. This is not widely applicable to all older adults.

In qualitative research, the role and position of the researcher naturally have an impact on the research performed, in both conducting and analysing research. The researchers who performed the interviews were of differing professions and ethnic backgrounds, which may have affected the way that participants responded to them, especially in cases where participants were of the same background as the interviewer, and so assumed knowledge or understanding of the cultural situation. A strength of our approach is the diversity of the research team. That similar findings came out of interviews conducted by each researcher shows that the questions used and the approach taken elicited these responses, rather than identification with the interviewer. Similarly, a diverse research team checked and ratified the analysis to minimise bias or oversight based on background or personal experience.

## CONCLUSIONS

BAME communities, including persons with dementia and their carers, in the UK have been heavily impacted by the COVID-19 pandemic. These impacts have been multifaceted, beginning with susceptibility to the virus and encompassing aspects of life as diverse and service access, social relationships and access to culturally specific resources. Underpinning many of these experiences has been a paucity of support. There are potentially long-reaching effects of these experiences that will be uncovered as the pandemic continues and beyond, including increased support needs of patients and carers, looking at promoting equity of service access and provision and exploring the role of technology in health and social care in the future. This paucity of support should be addressed to prevent potential widening inequalities and negative impacts on health and well-being.

**Acknowledgements** Thanks to our patient and public involvement panel for providing valuable input into study design and strategy.

**Contributors** PN performed the majority of interviews and developed the coding framework. EW performed extra interviews, contributed to the coding framework, performed analysis and drafted the manuscript. YB-M refined the coding framework and contributed to the manuscript. KRW, NK and ELS contributed to the manuscript. ND contributed to the manuscript and provided adjudication on analysis.

**Funding** This study/project is funded by the National Institute for Health Research (NIHR) School for Primary Care Research (project reference 489). The views expressed are those of the author(s) and not necessarily those of the NIHR or the Department of Health and Social Care. This work was supported by the Economic and Social Research Council [grant number: ES/V003720/1]. PN is funded through her NIHR In-Practice Fellowship (award number 300286) NK is supported by Alzheimer's Society Junior Fellowship grant funding (Grant Award number: 399 AS-JF-17b-016). ND is supported by Alzheimer's Society Junior Fellowship grant funding (Grant Award number: 399 AS-JF-16b-012). ELS are supported by Marie Curie core grant (number MCCC-FCO-16-U).

**Competing interests** None declared.

**Patient consent for publication** Not required.

**Ethics approval** The study was approved by UCL Research Ethics Committee (17623/002).

**Provenance and peer review** Not commissioned; externally peer reviewed.

**Data availability statement** Data are available upon reasonable request. Data can be provided in other forms if required. Please contact lead author for access.

**ORCID iDs**
Emily West http://orcid.org/0000-0003-3945-060X
Elizabeth L Sampson http://orcid.org/0000-0001-8929-7362
Nathan Davies http://orcid.org/0000-0001-7757-5353

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
