## [Reviewer comments · BMJ Open]

ARTICLE DETAILS

TITLE (PROVISIONAL)	An exploration of the impact of the Covid-19 pandemic on people with dementia and carers from Black and Minority Ethnic groups
AUTHORS	West, Emily; Nair, Pushpa; Barrado-Martin, Yolanda; Walters, Kate; Kupeli, Nuriye; Sampson, Elizabeth; Davies, Nathan

VERSION 1 – REVIEW

REVIEWER	Giebel, Clarissa University of Liverpool, Institute of Psychology, Health and Society
REVIEW RETURNED	12-Mar-2021

GENERAL COMMENTS	This is a really interesting and important study looking at the experiences of people with dementia and unpaid carers from a BAME background. the authors have included public involvement as well which is very good. I have very few comments about this paper, and it certainly adds an important piece of knowledge to the emerging evidence base. please see my comments below: - Need to define BAME acronym at first mention in Abstract.- The results in the abstract should be elaborated in a bit more detail, as opposed to just stating the identified themes.- Couple of grammar mistakes and missing words, i.e. this in the aims: "The aim of the study was to explore the effects of the Covid-19 pandemic ON people living with dementia"- The inclusion and exclusion criteria should not be listed like this in bullet points, as one does in an ethics protocol. these should be written in a paragraph.- The manuscript seems to have been hastily submitted, with various yellow highlighted sections in the methods and references missing, such as here: "This framework was further refined by EW. EW then iteratively coded transcripts using NVivo 11 (ref) according to the thematic framework, and in discussion with the wider research team. "- Please include age and gender in the description of the individual quotes, or, at least gender, as well as caring relationship.- There should be more quotes to illustrate the points made under 'Adapting to COVID-19', as there is currently only 1 very brief quote.- Again, grammar mistakes in Discussion, i.e.: "often-disadvantaged people"- There needs to be a greater focus on general impact of covid
--

	on dementia and the emerging evidence in the discussion, which is lacking. The findings need to be placed in context better, and compared to what other research has found in dementia in general, and highlight how this study adds a very important point of view, a BAME POV: https://www.ncbi.nlm.nih.gov/pmc/articles/PMC7929391/ ; https://bmcgeriatr.biomedcentral.com/articles/10.1186/s12877-020-01957-2 ; https://bmjopen.bmj.com/content/11/1/e045889
--	--

REVIEWER	Batool, Saqba The University of Manchester, Social Care and Society
REVIEW RETURNED	18-Mar-2021

GENERAL COMMENTS	This research contributes to the literature. It is undoubtedly an important and timely area of research. I believe the suggestions listed below could enhance the paper. These are:  1) The manuscript requires grammatical errors to be corrected. 2) More details about the samples' demographics would be beneficial. For example, how many of the 11 family carers and 4 people with dementia were south Asian and Afro-Caribbean. Also, details about their ages will be beneficial (e.g., were carers young or older carers as this can impact of practical implications) as well as what was participants first/preferred language. 3) Provide a breakdown of the south Asian and Afro-Caribbean participants. For example, south Asians are defined as individuals from Afghanistan, Pakistan, India, Nepal, Bhutan, Bangladesh, the Maldives and Sri Lanka so were individuals from all these backgrounds includes or some? 3) Provide details about the role of language and communication throughout the research process and what extra steps (if any) were taken to support participants. For instance, was language an issue in communicating, gaining consent and/or conducting interviews with the sample. If it was, how was this mitigated? 4) More details in relation to recruitment will be beneficial. For example, provide details on how participants were initially approached by local/national dementia/carer networks (e.g., where these individuals approached via telephone, letters via post, via regular communication they already had with the participants etc.?). Also, add what the process was in relation to the participants who were approached by the research team (e.g., how they were identified by the research team and approached). 5) Perhaps the wording in relation to geographic parameters could be worked on. The sample included individuals from in and around Greater London and further afield. Does this mean that they were from England or the UK with most of them being Greater London? 6) Provide some clarification on whether an inductive or deductive approach to data analysis was used. 7) Add a brief section that critically examines the role of the researcher on the participant and data collection that could
--

	have impacted on the research.
--	--------------------------------

VERSION 1 – AUTHOR RESPONSE

Reviewer 1 comments:		
Comment raised @	Response by author @	Location of revisions @
- Need to define BAME acronym at first mention in Abstract.	Thank you for your careful and considered feedback. This clarification has now been added to the manuscript.	Abstract, Introduction section
- The results in the abstract should be elaborated in a bit more detail, as opposed to just stating the identified themes.	Further detail has now been added.	Abstract
- Couple of grammar mistakes and missing words, i.e. this in the aims: "The aim of the study was to explore the effects of the Covid-19 pandemic ON people living with dementia"	Thank you for your thorough reading, the document has been thoroughly grammar checked by the team to check for other mistakes.	Entire manuscript
- The inclusion and exclusion criteria should not be listed like this in bullet points, as one does in an ethics protocol. these should be written in a paragraph.	These have now been amended into paragraphs	Research Design and Methods section
- The manuscript seems to have been hastily submitted, with various yellow highlighted sections in the methods and references missing, such as here: "This framework was further refined by EW. EW then iteratively coded transcripts using NVivo 11 (ref) according to the thematic framework, and in discussion with the wider research team. "	Apologies, highlighted sections have now been removed, and references checked.	Entire manuscript
- Please include age and gender in the description of the individual quotes, or, at least gender, as well as caring relationship.	As per advice from the editor (see beginning of this table), identifiers are limited. Age and gender have been added as identifiers throughout.	Throughout results section
- There should be more quotes to illustrate the points made under 'Adapting to COVID-19', as there is currently only 1 very brief quote.	More quotes have now been added to this section for deeper illustration of the theme.	Results section, "Adapting to COVID-19"
- Again, grammar mistakes in Discussion, i.e.: "often-disadvantaged people"	This compound adjective has been re-	Discussion

	written for clarity	
- There needs to be a greater focus on general impact of covid on dementia and the emerging evidence in the discussion, which is lacking. The findings need to be placed in context better, and compared to what other research has found in dementia in general, and highlight how this study adds a very important point of view, a BAME POV: https://www.ncbi.nlm.nih.gov/pmc/articles/PMC7929391/ ; https://bmcgeriatr.biomedcentral.com/articles/10.1186/s12877-020-01957-2 ; https://bmjopen.bmj.com/content/11/1/e045889	Thank you for your thoughtful suggestions, these have been incorporated into the discussion section, and the scope of this topic expanded	Discussion, final paragraph
Reviewer 2 comments:		
Comment raised @	Response by author @	Location of revisions @
1) The manuscript requires grammatical errors to be corrected.	Thank you for your thorough and thoughtful reading of the paper. The manuscript has been circulated to the research team for further checking throughout and correction of grammatical issues.	Entire manuscript
2) More details about the samples' demographics would be beneficial. For example, how many of the 11 family carers and 4 people with dementia were south Asian and Afro-Caribbean. Also, details about their ages will be beneficial (e.g., were carers young or older carers as this can impact of practical implications) as well as what was participants first/preferred language.	A table detailing demographic characteristics has now been added to the manuscript for clarity	Table 1
3) Provide a breakdown of the south Asian and Afro-Caribbean participants. For example, south Asians are defined as individuals from Afghanistan, Pakistan, India, Nepal, Bhutan, Bangladesh, the Maldives and Sri Lanka so were individuals from all these backgrounds includes or some?	This detail has now been included in the demographic table	Table 1
3) Provide details about the role of language and communication throughout the research process and what extra steps (if any) were taken to support participants. For instance, was language an issue in communicating, gaining consent and/or conducting interviews with the sample. If it was, how was this mitigated?	Further information on this issue has now been added to the data collection section to clarify this topic.	Research Design and Methods
4) More details in relation to recruitment will be beneficial. For example, provide details on how participants were initially approached by local/national dementia/carer	Further information on this issue has now been added	Research Design and

networks (e.g., where these individuals approached via telephone, letters via post, via regular communication they already had with the participants etc.?). Also, add what the process was in relation to the participants who were approached by the research team (e.g., how they were identified by the research team and approached).	to recruitment section to clarify this topic.	Methods
5) Perhaps the wording in relation to geographic parameters could be worked on. The sample included individuals from in and around Greater London and further afield. Does this mean that they were from England or the UK with most of them being Greater London?	Thank you for noticing this potential confusion – the phrasing has now been amended.	Research Design and Methods
6) Provide some clarification on whether an inductive or deductive approach to data analysis was used.	This information has now been added under the “Data Analysis” heading	Data Analysis
7) Add a brief section that critically examines the role of the researcher on the participant and data collection that could have impacted on the research.	Thank you for this valuable suggestion, this section has now been added in Strengths and Limitations	Discussion

VERSION 2 – REVIEW

REVIEWER	Giebel, Clarissa University of Liverpool, Institute of Psychology, Health and Society
REVIEW RETURNED	09-Apr-2021
GENERAL COMMENTS	It is difficult to make out where the authors have made changes, this should really be highlighted in red in the text and I'm unsure why the authors have not done this. The authors seem to have addressed the comments however.